# Influence of Pairing Startling Acoustic Stimuli with Postural Responses Induced by Light Touch Displacement

**John E. Misiaszek [1,2,\*], Sydney D. C. Chodan [1], Arden J. McMahon [1] and Keith K. Fenrich [1,2]**

[1] Department of Occupational Therapy, Faculty of Rehabilitation Medicine, University of Alberta, Edmonton, AB T6G 2G4, Canada; schodan@ualberta.ca (S.D.C.C.); ajmcmaho@ualberta.ca (A.J.M.); fenrich@ualberta.ca (K.K.F.)

[2] Neuroscience and Mental Health Institute, University of Alberta, Edmonton, AB T6G 2E1, Canada

\* Correspondence: john.misiaszek@ualberta.ca

**Abstract:** The first exposure to an unexpected, rapid displacement of a light touch reference induces a balance reaction in naïve participants, whereas an arm-tracking behaviour emerges with subsequent exposures. The sudden behaviour change suggests the first trial balance reaction arises from the startling nature of the unexpected stimulus. We investigated how touch-induced balance reactions interact with startling acoustic stimuli. Responses to light touch displacements were tested in 48 participants across six distinct combinations of touch displacement (DISPLACEMENT), acoustic startle (STARTLE), or combined (COMBINED) stimuli. The effect of COMBINED depended, in part, on the history of the preceding stimuli. Participants who received 10 DISPLACEMENT initially, produced facilitated arm-tracking responses with subsequent COMBINED. Participants who received 10 COMBINED initially, produced facilitated balance reactions, with arm-tracking failing to emerge until the acoustic stimuli were discontinued. Participants who received five DISPLACEMENT, after initially habituating to 10 STARTLE, demonstrated re-emergence of the balance reaction with the subsequent COMBINED. Responses evoked by light touch displacements are influenced by the startling nature of the stimulus, suggesting that the selection of a balance reaction to a threatening stimulus is labile and dependent, in part, on the context and sensory state at the time of the disturbance.

**Keywords:** light touch; balance; startle; human; standing

## 1. Introduction

Unexpected disturbances to balance are often met with whole-body reactions to stabilize the body and mitigate the potentially catastrophic consequences of a fall [1]. Moreover, the opportunity for generating a functionally meaningful response to a balance disturbance is normally limited to the initial exposure to the disturbance; rarely is there a second chance to get it right. Therefore, accurate interpretation of the sensory feedback related to balance disturbances is critical to generating appropriate balance reactions. Recently, rapid, unexpected displacements of a light touch reference were shown to evoke reactions consistent with a balance correction when standing with eyes closed, despite the absence of a mechanical disturbance to balance per se [2,3]. However, this putative balance reaction was not consistently expressed across participants and was only observed following the first unexpected displacement of the touch reference. With subsequent touch displacements, participants tracked the motion of the touch reference with a simple arm movement. This suggests that the sensation at the fingertip during the first trial was misinterpreted as a sway of the body away from the touch reference but was correctly interpreted as a displacement of the touch reference on subsequent trials.

The sudden change in behaviour between the first and subsequent exposures to the light touch displacement raises the possibility that the first trial response reflects a startle response. Startle reflexes are commonly defined as involuntary motor reactions to unexpected sensory stimuli that habituate with repeated exposure to the stimulus. Although startle reflexes are often described in relation to sudden auditory stimuli, startles can be elicited from a variety of sensory modalities, including tactile stimuli (reviewed in [4]). Moreover, it is well documented that the first exposure to unexpected balance disturbances are of larger amplitude than subsequent exposures to the same perturbations [5–9]. Campbell et al. [6] demonstrated that a large first trial response to a balance disturbance likely arises due to the superimposition of the balance reaction with a startle response, which subsequently habituates with repeated exposure to the balance disturbance. A characteristic feature of the habituation of balance reactions and startle responses is the attenuation of muscle activity associated with the evoked response. In contrast, the putative balance reactions to light touch displacement we previously reported [2,3] do not appear to habituate with subsequent exposures, but instead are replaced by a different behaviour within a single trial making the contribution of a startle response to this first trial reaction ambiguous.

Evidence from both human and animal studies have demonstrated that startle responses summate when evoked from more than one stimulus source [4]. Moreover, the summated responses are larger when startling stimuli of different modalities (for example, tactile and acoustic) are combined [10,11]. Blouin et al. [12] exploited this property of summation of startle reflexes to restore the amplitude of habituated postural responses in neck muscles, providing strong evidence that startle contributes to the larger postural responses observed with the initial disturbance. In this study, we aimed to determine what impact introducing a startling acoustic tone would have on the responses evoked by a rapid displacement of a light touch reference during standing. We hypothesized that, similar to our previous studies, an unexpected displacement of the light touch reference would result in a balance response on the first trial followed by arm-tracking behaviour on subsequent trials; however, subsequently combining an acoustic startle with the light touch displacement would promote re-expression of the balance response. We further hypothesized that initially combining an acoustic startle with the light touch displacement would result in a larger first trial balance response than with light touch displacement alone, but that the arm-tracking behaviour would emerge with habituation of the startle response. Finally, we hypothesized that allowing participants to habituate to the acoustic startle before introducing an unexpected displacement of the light touch reference would result in touch-evoked responses comparable to those observed in the absence of an accompanying acoustic startle.

## 2. Materials and Methods

Forty-eight healthy volunteers (29 female; 44 right-handed; median age 21 years; range 18–29 years) provided written consent to participate in a protocol performed in accordance with the Declaration of Helsinki, and approved by the University of Alberta Research Ethics Board (Pro00070448). It was essential that participants were unaware that the touch reference would be displaced or that startling tones would occur. As such, participants were screened to verify they were unaware of the study's protocol. Full disclosure of the study's purpose and procedures was provided after the experimental session and participants were provided the opportunity to withdraw their consent. One participant reported with nonsyndromic autosomal recessive hearing loss. However, post hoc review of their responses to the acoustic startle showed that they were indistinguishable from other participants and therefore the data were retained within the set. Otherwise, none of the participants reported any neurological or musculoskeletal disorders.

### 2.1. Set-Up and Apparatus

For all conditions, participants stood in stocking feet, shoulder width apart, on an ethylene-vinyl acetate (EVA) foam pad placed atop a 6 axis force plate (AMTI OR6-7-1000, Advanced Mechanical Technology Inc, Watertown, MA, USA) (Figure 1). During the test condition, participants were asked to touch a 3D-printed plastic touch plate mounted to a steel rod that permitted the height of the touch

plate to be adjusted to the participant's height. Participants were instructed to place the pad of the right index finger on a raised dimple (~0.5 mm) at the center of the touch plate, with the remaining fingers curled into the palm to avoid inadvertent contact with the plate. The use of the raised dimple was necessary as blind-folded participants will normally seek the edges of the touch plate. The height of the touch plate was adjusted so that participants could maintain contact of the finger pad with the wrist in a neutral position, the elbow flexed to approximately 90°, and a vertical alignment of the upper arm. During the No Touch conditions, the right arm was free to hang in a relaxed position at their side. The left arm was free to hang in a relaxed position at their side throughout the study. To produce a linear displacement of the touch plate, the plate was mounted on a square rail acme screw drive positioning stage (Lintech Positioning Systems 130 Series, Monrovia, CA, USA), driven by a computer-controlled two-phase stepper motor (Applied Motion Products 5023-124 2-phase hybrid stepper motor, Watsonville, CA, USA). The touch plate was displaced 12.5 mm, with a peak velocity of 124 mm/s. Stage position was measured using a linear displacement sensor (Penny & Giles SLS130, Penny & Giles Controls Limited, Christchurch, UK). The entire touch plate apparatus was on top of a 6 axis force plate (AMTI MC3A-100, Advanced Mechanical Technology, Inc., Watertown, MA, USA) to allow the vertical component of the touch force to be measured. The touch force was monitored online, and auditory feedback was provided if the force exceeded 1 N. Participants wore a pair of darkened goggles to block visual inputs.

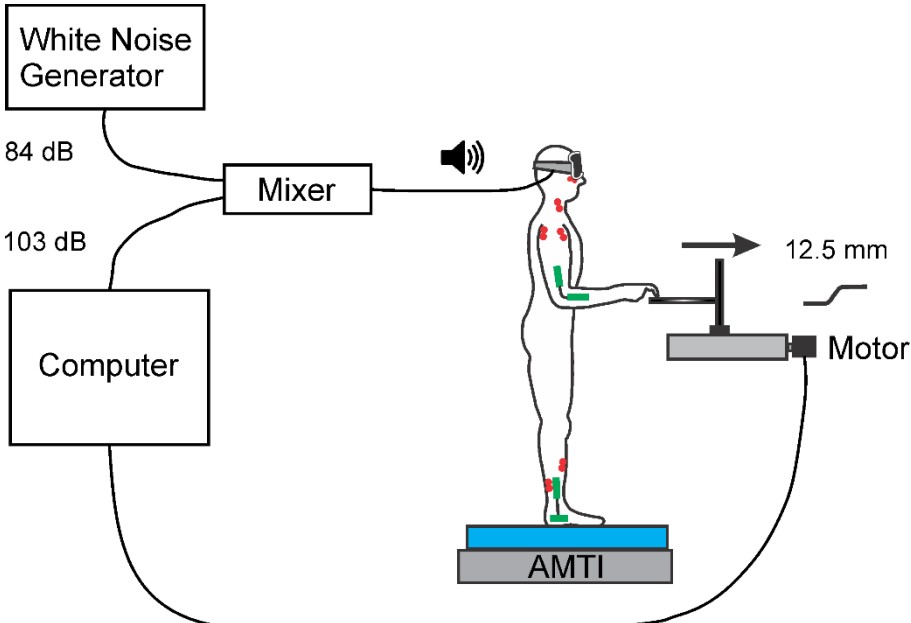

**Figure 1.** Schematic of the experimental set-up. Participants stood on foam atop a force plate, while lightly (<1N vertical load) touching a plastic touch plate. During DISPLACEMENT trials, the touch plate was unexpectedly translated away from the participant with the use of a computer controlled stepper motor. During STARTLE trials, the computer generated a startling tone delivered to the participant via a pair of indwelling earphones. During COMBINED trials, the touch displacement and startling tone were triggered simultaneously by the computer. In all conditions, the participant received white noise to mask the sound of the operating motor. Electromyographic (EMG) (red), goniometers (green), and center of pressure were used to characterize the evoked responses.

Auditory stimuli were delivered to the participants via a pair of Etymotic indwelling earphones (Etymotic ER-2, Etymotic Research Inc., Elk Grove Village, IL, USA). The startling acoustic stimulus (STARTLE) was generated from the PC sound card using a custom-written routine (LabView v8.2, National Instruments, Austin, TX, USA) and consisted of a 150 ms, 450 Hz tone delivered with a peak amplitude of 103 dB of sound pressure. Throughout the study, participants received white noise to

mask the sound of the operating motor and other extraneous sounds. The white noise was calibrated at its maximum level and had a peak amplitude of 84 dB. Therefore, the minimum STARTLE signal to white noise was 19 dB of sound pressure. The STARTLE and white noise were calibrated using an Audioscan Verifit (Audioscan, Dorchester, ON, Canada) fitted with a 2-cc coupler. The STARTLE and white noise were mixed (Shure SCM268 Mixer, Shure Inc., Niles, IL, USA) prior to delivery to the earphones.

### 2.2. Protocol

Participants were randomly allocated to one of six cohorts. Each cohort represented one of the six possible sequences of the stimulus presentation orders for the three stimulus types: touch displacement alone (DISPLACEMENT), STARTLE, or simultaneous presentation of DISPLACEMENT and STARTLE (COMBINED). Pilot testing indicated that DISPLACEMENT and STARTLE evoked responses in tibialis anterior (TA) and anterior deltoid (AD) with comparable latency, but that STARTLE yielded shorter latency responses in sternocleidomastoid (SCM) and orbicularis oculi (OO) (also see Figure 8). Consequently, COMBINED stimuli were presented with a 0 ms lag (i.e., synchronized delivery via the computerized control program and verified post hoc from the linear displacement and acoustic tone recordings), given that the primary objective of this study was the interaction of STARTLE with DISPLACEMENT on responses related to activity in TA or AD. Each cohort completed four conditions: (1) standing eyes open, (2) standing eyes open with light touch, (3) standing eyes closed, and (4) the test condition. During the test condition participants were asked to stand with eyes closed with light touch. Approximately 10 s into the test condition the first stimulus trial was delivered. An additional 9 stimuli of that type were delivered, followed by 5 each of the other two stimuli types, for a total of 20 stimuli. Stimuli were separated by at least 8 s intervals. The test condition took up to 7 min to complete. Conditions 1 to 3 were 90 s each and were performed to create the expectation that the test condition would be uneventful. Participants rested for 2 min between conditions.

### 2.3. Recording and Data Acquisition

Electromyographic (EMG) activity was recorded from TA and soleus (SOL) of the right leg; AD and posterior deltoid (PD) of the right arm; and the right SCM and OO. EMG activity was recorded using pairs of Ag/AgCl electrodes (NeuroPlus A10040, Vermed, Bellows Falls, VT, USA) placed on the skin over the bellies of the intended muscles, with an inter-electrode distance of about 2 cm. Ground electrodes were placed over the right clavicle and the anterior tibia of the right leg. The skin at the limb EMG recording sites was shaved with a razor and cleaned with alcohol, while the skin below the eye and at the neck was only cleaned with alcohol. Electrode impedance was less than 20 kΩ at all recording sites (Grass F-EZM5 impedance meter, Astro-Med, Inc., West Warwick, Rhode Island, USA). The EMG signals were amplified and band-pass filtered (10 Hz-1 kHz with a 60 Hz notch filter, Grass P511 amplifiers, Astro-Med, Inc., West Warwick, RI, USA) prior to digitization. Electrogoniometers were placed across the right ankle (SG110A, Biometrics Ltd., Newport, UK) and elbow joints (SG110, Biometrics Ltd., Newport, UK). All analog signals were digitized at 2000 Hz (PCI-MIO-16E-4, National Instruments, Austin, TX, USA) and stored directly to hard drive using a custom-written LabView data acquisition routine.

### 2.4. Data Analysis

Post-processing of the signals was performed offline using custom-written LabView routines. The EMG signals were digitally full-wave rectified and then low-pass filtered (50 Hz, 4th order zero-lag Butterworth filter). The mechanical signals were low-pass filtered (20 Hz, 2nd order zero-lag Butterworth filter) and the position of the center of pressure was calculated from the force and moment signals from the force plate. For each stimulus, a 900 ms trace was extracted from the continuous data feed, aligned to the onset of the stimulus and included a 300 ms pre-stimulus period.

To determine whether the stimulus evoked an EMG response in the recorded muscles, a two standard deviations band around the mean EMG activity for the 100 ms prior to the perturbation onset was calculated. A response was considered to be present if following the onset of the stimulus the EMG trace exceeded this band for at least 20 continuous milliseconds. The onset latency of an evoked response was taken as the time following the stimulus onset that the EMG trace first exceeded the two standard deviations band. For consistency with our previous studies [2,3], only responses with onset latencies <200 ms were considered.

A key outcome measure of interest was the habituation of EMG response amplitudes with successive presentations of a stimulus. Therefore, it was important to estimate a response amplitude even in cases when the criteria for a response (described in the previous paragraph) were not met as it is possible that small amplitude responses were not identified. Consequently, EMG amplitude was calculated for each stimulus trial. To do so, analysis windows were established for each muscle by overlaying all 20 of the individual traces for that participant and manually placing cursors to capture the onset and offset of the apparent initial evoked response. This was preferred to using the average onset latency as average onset latencies will cleave the initial rise in a response in some of the trials due to the natural trial by trial variation or when the onset latency was different between stimulus types (for example, in OO and SCM, see Results and Figure 2). EMG response amplitudes were normalized to the maximum voluntary contraction obtained at the end of the experimental session.

As noted previously [3], participants sway considerably when standing on foam, making the two standard deviations method to identify stimulus-evoked events impractical for the mechanical signals. Consequently, the change in anterior–posterior position of the center of pressure (COP$_{AP}$) was calculated as the difference in position 300 ms following the stimulus onset, relative to the position at stimulus onset, for all trials. Changes in elbow and ankle angles were calculated using the same approach.

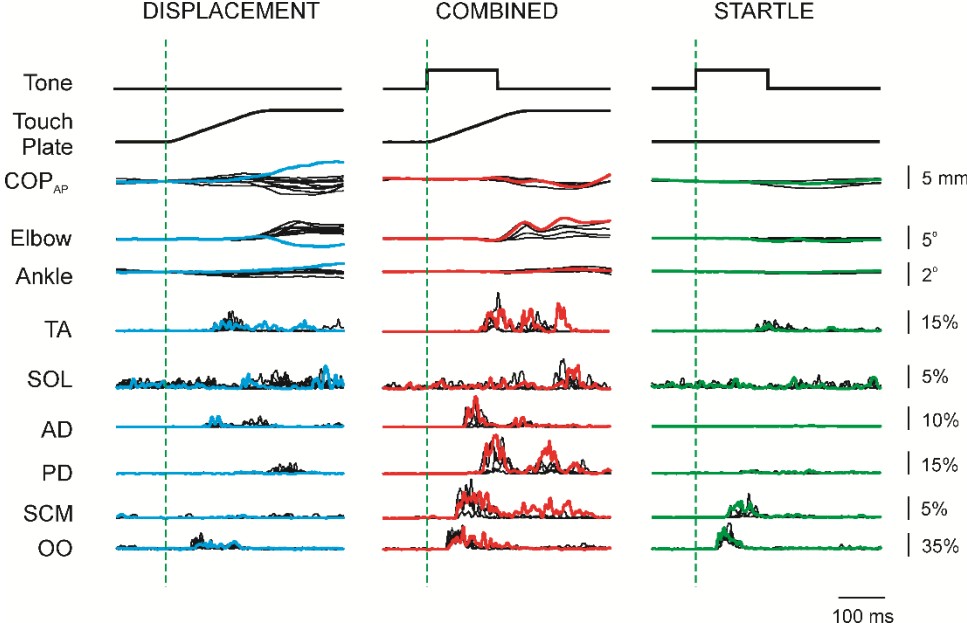

**Figure 2.** Sample data traces from a single participant who received the COMBINED stimuli after habituating to the DISPLACEMENT stimuli. Each cluster of traces represents the complete series of trials for that stimulus mode, with the coloured traces representing the first trial for that stimulus mode (Blue, DISPLACEMENT; Red, COMBINED; Green, STARTLE). The thin black lines of each cluster represent the subsequent trials. The vertical dashed line is aligned to the onset of the stimuli. Positive deflections in the COP$_{AP}$ traces represent forward, elbow traces represent extension, and ankle traces represent plantarflexion. COP$_{AP}$, center of pressure anterior–posterior; TA, tibialis anterior; SOL, soleus; AD, anterior deltoid; PD, posterior deltoid; SCM, sternocleidomastoid; OO, orbicularis oculi.

*2.5. Statistics*

Amplitude of evoked EMG responses or mechanical events were compared across repeated trials within a protocol cohort using one-way repeated measures analyses of variance (rmANOVA). Six levels were used for each rmANOVA: Trial 1, the mean of Trials 8–10, Trial 11, Trial 15, Trial 16 and Trial 20. The mean of Trials 8–10 was used to represent the habituated response to the initial stimulus, whereas Trials 1, 11, 15, 16 and 20 represent the transition points between stimulus types. Significant main effects were then assessed using Tukey's Honest Significant Difference (HSD) tests, but only immediately adjacent points in the sequence were compared (e.g., Trial 1 vs. Trials 8–10, Trials 8–10 vs. Trial 11 etc.).

Note that for each EMG response or mechanical event, only participants with complete datasets for that outcome measure were included for analysis. Thus, if an EMG response was contaminated, such as by unexpected activity prior to the stimulus onset, then the data from that participant for that muscle were excluded. EMG data were also excluded on three occasions because of persistent noise in the recording that could not be eliminated. Due to a technical issue with the electrogoniometers, the elbow and ankle recordings were not included in the datasets of 10 and 5 participants, respectively. Additional participants were recruited to ensure a minimum of 5 complete datasets were available for each outcome measure for each cohort, while maintaining balanced numbers in all cohorts. Doing so brought the total number of participants in each cohort to 8. Consequently, the number of datasets for each outcome measure varied between 5 and 8 across the cohorts. This is reflected in the *n* reported in the Tables of statistics presented in the Results.

A comparison of EMG and mechanical response amplitudes was also performed between the DISPLACEMENT and COMBINED stimuli for those participants who received these stimuli during the initial 10 trial sequence of stimuli. Doing so allowed for up to 16 participants to be included in each group by combining the participants from the two cohorts receiving each initial stimulus type. For this analysis, a two-way ANOVA with one repeated factor was employed. Stimulus type (DISPLACEMENT vs. COMBINED) was the independent factor, while Trial (Trial 1 vs. mean of Trials 8–10) was the repeated factor. Tukey's HSD tests were used to assess the nature of significant interaction terms or main effects.

EMG response onset latencies were compared only for trials with a demonstrable evoked response. Therefore, the dataset of 20 trials for any given participant was rarely complete. Indeed, no observable responses were identified in one or more muscles of some participants. As the primary objective of the latency comparison was to assess the effect of combining the displacement with the acoustic startle, we took the average onset latency across all trials for a given stimulus type for each participant for each muscle. In this way, each participant contributed a single value for each stimulus type to be considered in the analysis. If a participant did not express at least one response for each stimulus type then that participant was excluded from the analysis for that muscle. A one-way rmANOVA was then employed, with 3 levels for stimulus type (DISPLACEMENT, STARTLE, COMBINED).

Statistical analyses were performed with the *Real Statistics Using Excel* Resource Pack software (Release 6.2). Copyright (2013–2019) Charles Zaiontz [13]. All comparisons were made with $\alpha = 0.05$. Descriptive statistics are presented as the mean ± standard error of the mean (SE). The complete dataset, along with participant characteristics, is available in the supplemental material (Spreadsheet S1).

## 3. Results

*3.1. Superimposition of Acoustic Startle on Habituated Touch Displacement Responses*

Unexpected touch displacements evoked responses comparable to what has been reported previously [2,3]. The leftmost column of traces in Figure 2 depicts data from one participant who displayed a forward sway reaction (COP$_{AP}$) coupled with an elbow flexion after the initial unexpected touch displacement (blue traces), but adapted to an arm-tracking behaviour (elbow extension) with minimal forward sway after the subsequent displacements (black traces). STARTLE was superimposed

with the touch displacement for the subsequent five trials, displayed in the middle column of traces in Figure 2. For this participant, the initial COMBINED stimulus (red traces) resulted in the continued expression of the arm-tracking behaviour, but with larger amplitude EMG responses, or emergence of additional EMG responses, compared with the DISPLACEMENT responses. For the final five trials, STARTLE was presented alone (rightmost column), which yielded distinct bursts of EMG activity in TA, SCM and OO, but minimal impact on the $COP_{AP}$ position or elbow angle, relative to the preceding DISPLACEMENT or COMBINED stimuli.

The graphs in Figure 3A display average data from all participants who received the sequence of stimuli shown in Figure 2. As shown in the $COP_{AP}$ and elbow angle graphs, the initial unexpected touch displacement resulted in large forward sway (Trial 1 = 8.1 ± 3.4 mm), coupled with a distinct elbow flexion (Trial 1 = 3.2 ± 1.4°). On subsequent touch displacement trials, the forward sway was substantially reduced, and the elbow switched to an extension response. By the last three trials of DISPLACEMENT, the forward sway was minimal (Trials 8–10 = 0.5 ± 0.3 mm) and the elbow extension was consistent (Trials 8–10 = 2.6 ± 0.6°). With the superimposition of STARTLE on the touch displacement elbow extension was markedly increased (Trial 11 = 7.5 ± 1.9°), but $COP_{AP}$ sway was not evident (Trial 11 = –0.2 ± 1.6 mm). Subsequently, the augmented elbow extension abated (Trial 15 = 3.6 ± 1.0°), comparable to the extension observed at the end of the DISPLACEMENT trials. The subsequent STARTLE alone trials did not evoke responses in these two metrics. EMG response amplitudes are also shown for TA, AD, SCM and OO. The initial DISPLACEMENT resulted in a burst of TA activity (Trial 1 = 7.6 ± 3.2% MVC), but not AD (Trial 1 = 0.6 ± 0.3% MVC), consistent with a forward sway response. With subsequent DISPLACEMENT trials the TA burst decreased (Trials 8–10 = 0.2 ± 0.2% MVC), while AD demonstrated a clear burst (Trials 8–10 = 3.9 ± 0.7% MVC), consistent with the occurrence of the arm-tracking behaviour. EMG activity in SCM or OO was not consistently expressed with DISPLACEMENT alone. Superimposition of STARTLE and DISPLACEMENT resulted in large bursts of EMG activity in all four muscles displayed. The first COMBINED trial resulted in a large burst in TA (Trial 11 = 21.9 ± 8.5% MVC) that abated with repeated trials (Trial 15 = 8.0 ± 3.1% MVC). Similar effects were observed in AD (Trial 11 = 18.4 ± 5.7; Trial 15 = 8.9 ± 3.2% MVC) and SCM (Trial 11 = 42.1 ± 15.0% MVC; Trial 15 = 15.0 ± 6.0% MVC), but not OO which displayed a burst amplitude unaffected by the repetition of the stimulus (Trial 11 = 21.7 ± 8.4% MVC; Trial 15 = 30.7 ± 5.7% MVC). Subsequently, with the STARTLE alone, bursts in TA and AD were markedly reduced (Trial 16 = 2.0 ± 1.1% MVC and 1.9 ± 1.5% MVC, respectively), whereas bursts in SCM continued to progressively decline with continued stimuli. In contrast, burst amplitudes in OO continued to be strongly expressed and did not appear to habituate with repeated STARTLE. A complete report of the descriptive statistics and the result of the associated ANOVAs is provided in Table 1A.

Figure 3B displays data from participants who were first habituated to DISPLACEMENT and then exposed to the STARTLE alone, followed by COMBINED stimuli. As can be seen, the DISPLACEMENT data for this cohort of participants largely replicates the results for the cohort shown in Figure 3A. Similarly, STARTLE resulted in minimal changes to $COP_{AP}$ or elbow angle. STARTLE also evoked minimal activity in TA and AD. In contrast, STARTLE initially evoked a pronounced burst of activity in SCM (Trial 11 = 20.5 ± 6.1% MVC) that rapidly decreased with repeated STARTLE trials (Trial 15 = 6.1 ± 2.7% MVC), whereas the large initial burst in OO (Trial 11 = 13.7 ± 4.4% MVC) was not influenced by repeated STARTLE trials (Trial 15 = 11.4 ± 4.7% MVC). With the STARTLE superimposed on the DISPLACEMENT, the arm-tracking behaviour was expressed with an elbow extension (Trial 16 = 4.2 ± 0.7°), which remained unchanged with repeated COMBINED trials (Trial 20 = 3.2 ± 0.4°). COMBINED stimuli evoked inconsistent, small bursts of activity in TA, whereas bursts in AD (Trial 16 = 4.4 ± 1.8% MVC) were comparable to the bursts expressed with DISPLACEMENT and were relatively stable in amplitude with repeated trials (Trial 20 = 3.8 ± 0.9% MVC). Activity in SCM with the COMBINED stimuli was not different from the habituated STARTLE response. OO burst amplitude with the COMBINED stimuli remained similar to the STARTLE response

and did not change with repeated trials. A complete report of the descriptive statistics and the result of the associated ANOVAs is provided in Table 1B.

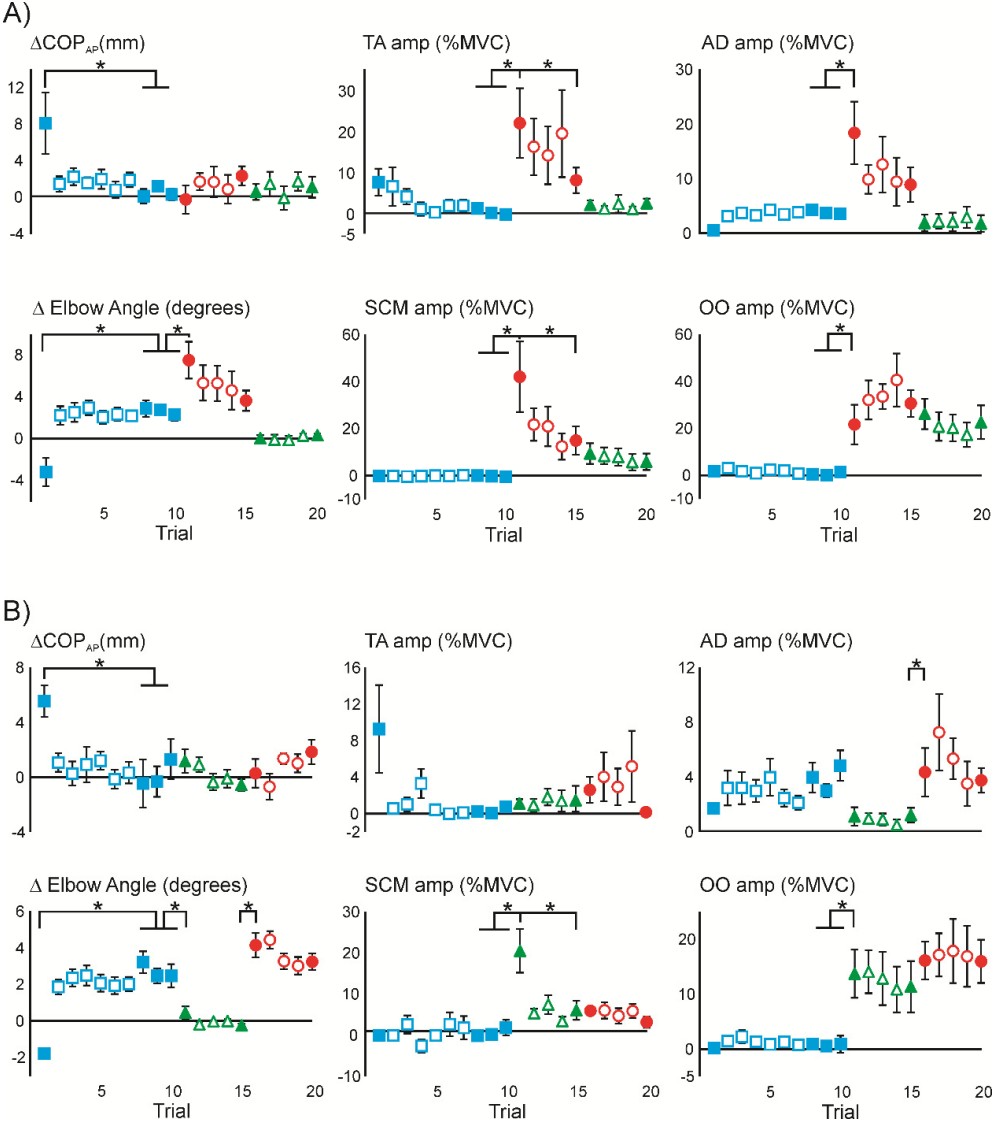

**Figure 3.** Trial-by-trial mean (SE) mechanical and muscle response amplitudes for participants who habituated to the DISPLACEMENT stimuli initially. (**A**) Participants subsequently received the COMBINED stimuli, followed by the STARTLE. (**B**) Participants subsequently received the STARTLE, followed by the COMBINED stimuli. Asterisks indicate differences identified by Tukey's HSD comparisons ($p < 0.05$). Blue squares, DISPLACEMENT; Red circles, COMBINED; Green triangles, STARTLE. Closed symbols indicate data points used in the statistical analysis. TA, tibialis anterior; AD, anterior deltoid; SCM, sternocleidomastoid; OO, orbicularis oculi; $COP_{AP}$, center of pressure anterior–posterior; MVC, maximum voluntary contraction.

### 3.2. Initial Superimposition of Acoustic Startle and Touch Displacement

Presenting naïve participants with the STARTLE superimposed on DISPLACEMENT in the first 10 trials evoked pronounced forward sway, coupled with elbow flexion in the initial trial (Figure 4, red traces). Subsequent exposures to the COMBINED stimuli continued to evoke a forward sway with elbow flexion (black traces), but of smaller amplitude. The participant displayed in Figure 4 continued to express a modest elbow flexion throughout the 10 COMBINED trials and never adopted an arm-tracking behaviour. Subsequent STARTLE trials did not evoke a discernible response in this

participant, outside of a persistent blink reflex (OO). The DISPLACEMENT trials at the end of the sequence of stimuli resulted in a modest elbow extension on the first exposure (blue traces), that persisted thereafter (black traces) in this participant.

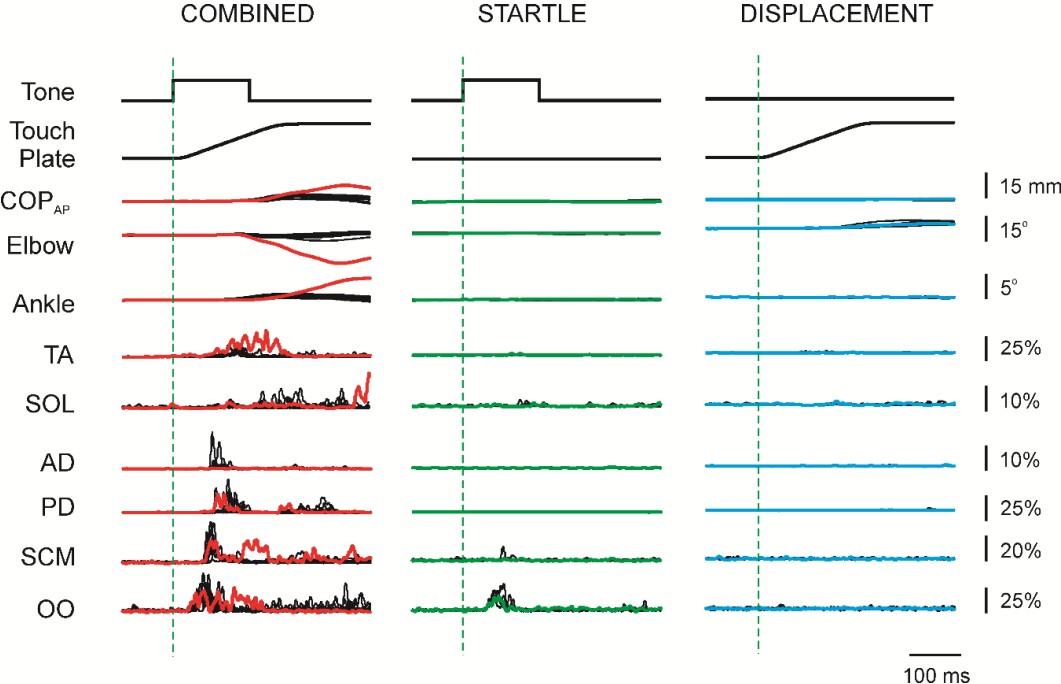

**Figure 4.** Sample data traces from a single participant who received the COMBINED stimuli initially. Each cluster of traces represents the complete series of trials for that stimulus mode, with the coloured traces representing the first trial for that stimulus mode (Red, COMBINED; Green, STARTLE; Blue, DISPLACEMENT). The thin black lines of each cluster represent the subsequent trials. The vertical dashed line is aligned to the onset of the stimuli. Positive deflections in the COP$_{AP}$ traces represent forward, elbow traces represent extension, and ankle traces represent plantarflexion. COP$_{AP}$, center of pressure anterior–posterior; TA, tibialis anterior; SOL, soleus; AD, anterior deltoid; PD, posterior deltoid; SCM, sternocleidomastoid; OO, orbicularis oculi.

Group averaged data for the cohorts that received the COMBINED stimulus initially are displayed in Figure 5. Figure 5A depicts the cohort that received the same sequence of stimuli as the participant in Figure 4. As can be seen, the initial pronounced forward sway (Trial 1 = 18.8 ± 6.0 mm) and elbow flexion (Trial 1 = 17.1 ± 3.3°) was a consistent outcome for all participants who received the COMBINED stimulus in the first trial. Subsequently, the forward sway persisted, but decreased (Trials 8–10 = 5.9 ± 2.7 mm), while the elbow flexion decreased towards no discernible response (Trials 8–10 = 0.8 ± 0.5° of flexion). Pronounced EMG bursts were evoked in each of the muscles depicted with the initial COMBINED stimuli. Where the response amplitudes in TA and SCM decreased by 2/3 of their first trial amplitude on average, AD burst amplitude decreased more modestly and OO burst amplitudes were unchanged with repeated exposure to the COMBINED stimuli. The subsequent STARTLE alone trials evoked a blink reflex in OO that was not different in amplitude from the response evoked in the preceding COMBINED trials, but little else of note in the other measures. Thereafter, the DISPLACEMENT trials resulted in modest forward sway (Trial 16 = 3.0 ± 2.1 mm) that decreased with repeated trials (Trial 20 = 0.8 ± 1.0 mm), and a minimal elbow extension (Trial 16 = 0.7 ± 0.4°) that persisted with repeated trials (Trial 20 = 1.0 ± 0.3°). Small, persistent bursts were observed in AD, while little activity in TA, SCM or OO was evoked with DISPLACEMENT trials. A complete report of the descriptive statistics and the result of the associated ANOVAs is provided in Table 2A.

**Table 1.** Change in $COP_{AP}$ position and elbow angle, and normalized muscle response amplitudes.

| | Stimulus | | | | | | | Statistical Results | |
|---|---|---|---|---|---|---|---|---|---|
| A | $D_1$ | $D_{8-10}$ | $C_1$ | $C_5$ | $S_1$ | $S_5$ | $n$ | Main Effect | Tukey's HSD ($p < 0.05$) |
| **Mechanical** | | | | | | | | | |
| $\Delta COP_{AP}$ (mm) | 8.1 (3.4) | 0.5 (0.3) | −0.2 (1.6) | 2.3 (1.1) | 0.6 (0.9) | 1.1 (1.2) | 8 | 3.38 ($p = 0.014$) | $D_1 > D_{8-10}$ |
| $\Delta$Elbow (deg) | −3.2 (1.4) | 2.6 (0.6) | 7.5 (1.9) | 3.6 (1.0) | −0.3 (0.3) | 0.0 (0.2) | 7 | 11.6 ($p = 0.000$) | $D_1 < D_{8-10}$, $D_{8-10} < C_1$ |
| | | | | | | | | | |
| **EMG (%MVC)** | | | | | | | | | |
| TA | 7.6 (3.3) | 0.2 (0.2) | 21.9 (8.5) | 8.0 (3.1) | 2.0 (1.1) | 2.4 (1.1) | 8 | 6.31 ($p = 0.000$) | $D_{8-10} < C_1$, $C_1 > C_5$ |
| AD | 0.7 (0.3) | 3.9 (0.7) | 18.4 (5.7) | 8.9 (3.2) | 1.9 (1.5) | 1.8 (1.6) | 8 | 7.24 ($p = 0.000$) | $D_{8-10} < C_1$ |
| SCM | −0.2 (0.2) | −0.2 (0.1) | 42.1 (15.0) | 15.0 (6.0) | 9.4 (4.4) | 6.0 (3.5) | 8 | 6.74 ($p = 0.000$) | $D_{8-10} < C_1$, $C_1 > C_5$ |
| OO | 1.8 (1.5) | 0.7 (0.4) | 21.7 (8.4) | 30.7 (5.7) | 26.2 (6.4) | 22.7 (7.2) | 8 | 9.71 ($p = 0.000$) | $D_{8-10} < C_1$ |
| | Stimulus | | | | | | | Statistical Results | |
| B | $D_1$ | $D_{8-10}$ | $S_1$ | $S_5$ | $C_1$ | $C_5$ | $n$ | Main Effect | Tukey's HSD ($p < 0.05$) |
| **Mechanical** | | | | | | | | | |
| $\Delta COP_{AP}$ (mm) | 5.5 (1.2) | 0.2 (1.0) | 2.8 (0.9) | −0.6 (0.4) | 0.3 (1.0) | 1.8 (0.9) | 8 | 7.79 ($p = 0.000$) | $D_1 > D_{8-10}$ |
| $\Delta$Elbow (deg) | −1.8 (0.3) | 2.7 (0.3) | 0.5 (0.3) | −0.2 (0.1) | 4.2 (0.7) | 3.2 (0.4) | 5 | 29.4 ($p = 0.000$) | $D_1 < D_{8-10}$, $D_{8-10} > S_1$, $S_5 < C_1$ |
| | | | | | | | | | |
| **EMG (%MVC)** | | | | | | | | | |
| TA | 9.3 (4.8) | 0.4 (0.3) | 1.1 (0.5) | 1.6 (1.4) | 2.6 (1.4) | 0.2 (0.2) | 8 | 2.50 ($p = 0.049$) | |
| AD | 1.7 (0.4) | 4.0 (0.7) | 1.1 (0.7) | 1.2 (0.5) | 4.4 (1.8) | 3.8 (0.9) | 8 | 4.12 ($p = 0.005$) | $S_5 < C_1$ |
| SCM | −0.1 (0.2) | 0.6 (0.8) | 20.5 (6.1) | 6.1 (2.7) | 5.9 (0.6) | 3.2 (1.6) | 6 | 6.95 ($p = 0.000$) | $D_{8-10} < S_1$, $S_1 > S_5$ |
| OO | 0.1 (0.3) | 0.8 (0.7) | 13.7 (4.4) | 11.4 (4.7) | 16.2 (3.4) | 16.0 (3.9) | 8 | 12.1 ($p = 0.000$) | $D_{8-10} < S_1$ |

Values are means (SE), $COP_{AP}$, center of pressure anterior–posterior; TA, tibialis anterior; AD, anterior deltoid; SCM, sternocleidomastoid; OO, orbicularis oculi; D, touch displacement; S, startle; C, combined; MVC, maximum voluntary contraction. Statistical results are $F$ statistics and $p$ values. Tukey's HSD applied to adjacent points in sequence only.

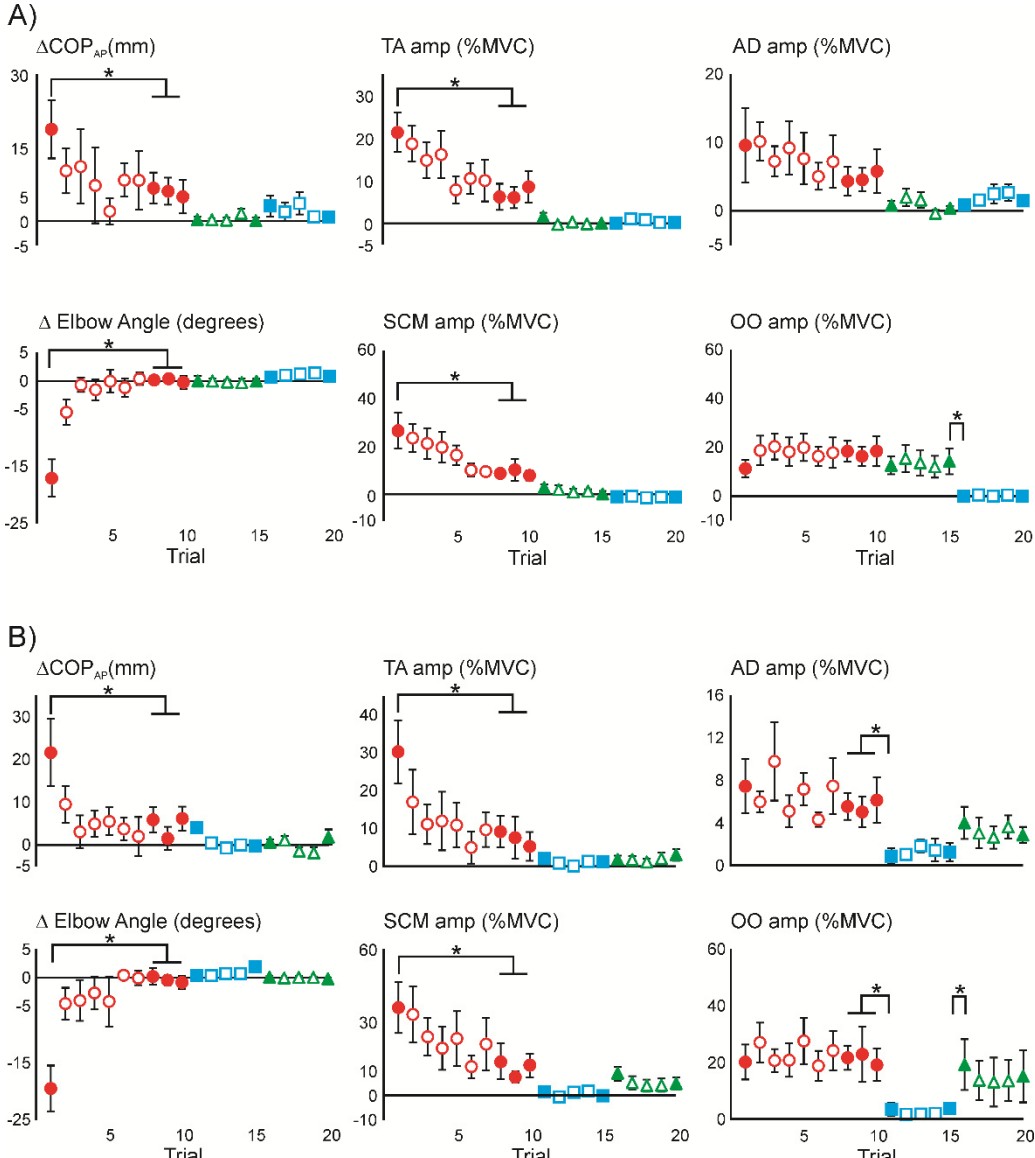

**Figure 5.** Trial-by-trial mean (SE) mechanical and muscle response amplitudes for participants who habituated to the COMBINED stimuli initially. (**A**) Participants subsequently received the STARTLE stimuli, followed by the DISPLACEMENT. (**B**) Participants subsequently received the DISPLACEMENT, followed by the STARTLE. Asterisks indicate differences identified by Tukey's HSD comparisons ($p < 0.05$). Blue squares, DISPLACEMENT; Red circles, COMBINED; Green triangles, STARTLE. Closed symbols indicate data points used in the statistical analysis. TA, tibialis anterior; AD, anterior deltoid; SCM, sternocleidomastoid; OO, orbicularis oculi; $COP_{AP}$, center of pressure anterior–posterior; MVC, maximum voluntary contraction.

Figure 5B displays the averaged data for the cohort of participants who received the COMBINED stimuli initially, followed by DISPLACEMENT and then STARTLE alone. The COMBINED stimuli data for this cohort are qualitatively similar to what is shown in Figure 5A. Of note, the COMBINED stimuli evoked a pronounced forward sway (Trial 1 = 21.6 ± 7.9 mm) that decreased with repeated trials (Trials 8–10 = 4.5 ± 2.4 mm), concomitantly with a pronounced initial elbow flexion (Trial 1 = 19.5 ± 4.0°) that progressively decreased until no response at the elbow was apparent (Trials 8–10 = 0.3 ± 0.9° of flexion on average). Subsequently, the data resulting from DISPLACEMENT and STARTLE are similar to the comparable stimuli of Figure 5A. A complete report of the descriptive statistics and the result of the associated ANOVAs is provided in Table 2B.

**Table 2.** Change in $COP_{AP}$ position and elbow angle, and normalized muscle response amplitudes.

| | Stimulus | | | | | | | Statistical Results | |
|---|---|---|---|---|---|---|---|---|---|
| A | $C_1$ | $C_{8-10}$ | $S_1$ | $S_5$ | $D_1$ | $D_5$ | $n$ | Main Effect | Tukey's HSD ($p < 0.05$) |
| **Mechanical** | | | | | | | | | |
| $\Delta COP_{AP}$ (mm) | 18.8 (6.0) | 5.9 (2.7) | 0.3 (1.0) | 0.0 (0.7) | 3.0 (2.1) | 0.8 (1.0) | 8 | 6.98 ($p = 0.000$) | $C_1 > C_{8-10}$ |
| $\Delta$Elbow (deg) | −17.1 (3.3) | −0.8 (0.5) | 0.0 (0.1) | 0.0 (0.1) | 0.7 (0.4) | 1.0 (0.3) | 8 | 27.1 ($p = 0.000$) | $C_1 > C_{8-10}$ |
| | | | | | | | | | |
| **EMG (%MVC)** | | | | | | | | | |
| TA | 21.3 (4.6) | 7.0 (2.5) | 1.7 (0.9) | 0.1 (0.1) | 0.1 (0.2) | 0.2 (0.1) | 8 | 17.7 ($p = 0.000$) | $C_1 > C_{8-10}$ |
| AD | 9.6 (5.5) | 4.9 (2.1) | 0.8 (0.7) | 0.4 (0.3) | 0.8 (0.6) | 1.5 (0.8) | 8 | 2.88 ($p = 0.028$) | |
| SCM | 27.2 (7.8) | 9.8 (2.0) | 3.7 (1.5) | 1.1 (1.4) | 0.0 (0.1) | −0.1 (0.4) | 7 | 11.5 ($p = 0.000$) | $C_1 > C_{8-10}$ |
| OO | 11.3 (3.5) | 17.8 (4.4) | 12.6 (3.6) | 14.3 (5.3) | 0.0 (0.5) | 0.0 (0.3) | 8 | 9.09 ($p = 0.000$) | $S_5 > D_1$ |
| | Stimulus | | | | | | | Statistical Results | |
| B | $C_1$ | $C_{8-10}$ | $D_1$ | $D_5$ | $S_1$ | $S_5$ | $n$ | Main Effect | Tukey's HSD ($p < 0.05$) |
| **Mechanical** | | | | | | | | | |
| $\Delta COP_{AP}$ (mm) | 21.6 (7.9) | 4.5 (2.4) | 4.1 (1.3) | -0.2 (0.5) | 0.6 (0.7) | 1.8 (1.8) | 8 | 5.99 ($p = 0.000$) | $C_1 > C_{8-10}$ |
| $\Delta$Elbow (deg) | −19.5 (4.0) | −0.3(0.8) | 0.4 (0.2) | 1.9 (0.7) | 0.0 (0.3) | -0.2 (0.2) | 8 | 24.1 ($p = 0.000$) | $C_1 > C_{8-10}$ |
| | | | | | | | | | |
| **EMG (%MVC)** | | | | | | | | | |
| TA | 30.3 (8.3) | 7.3 (4.4) | 2.1 (1.4) | 1.3 (0.9) | 1.8 (1.0) | 3.1 (1.5) | 8 | 10.1 ($p = 0.000$) | $C_1 > C_{8-10}$ |
| AD | 7.4 (2.6) | 5.6 (1.3) | 0.9 (0.7) | 1.2 (0.9) | 4.0 (1.5) | 2.9 (0.8) | 8 | 4.32 ($p = 0.004$) | $C_{8-10} > D_1$ |
| SCM | 36.0 (11.3) | 11.2 (3.7) | 1.3 (1.2) | −0.3 (0.2) | 8.8 (3.1) | 4.7 (2.6) | 7 | 7.58 ($p = 0.000$) | $C_1 > C_{8-10}$ |
| OO | 20.2 (6.1) | 21.3 (6.2) | 3.7 (2.3) | 4.0 (2.0) | 19.3 (9.0) | 15.2 (9.2) | 8 | 4.10 ($p = 0.005$) | $C_{8-10} > D_1$, $D_5 < S_1$ |

Values are means (SE), $COP_{AP}$, center of pressure anterior–posterior; TA, tibialis anterior; AD, anterior deltoid; SCM, sternocleidomastoid; OO, orbicularis oculi; D, touch displacement; S, startle; C, combined; MVC, maximum voluntary contraction. Statistical results are *F* statistics and *p* values. Tukey's HSD applied to adjacent points in sequence only.

### 3.3. Comparing Initial COMBINED with Initial DISPLACEMENT

To compare the effect of superimposing STARTLE with DISPLACEMENT with the responses observed with DISPLACEMENT alone in the initial trials of naïve participants, the two cohorts of each stimulus type were collapsed. The group averaged data comparing the initial COMBINED stimuli with the initial DISPLACEMENT stimuli are presented in Figure 6. As seen in the $COP_{AP}$ data, COMBINED resulted in a generally larger forward sway response than for DISPLACEMENT. The ANOVA indicated a significant Stimulus x Trial interaction ($F_{130} = 4.44$, $p = 0.04$), as the sway amplitude converged with the later trials. The first trial sway response was significantly larger for the COMBINED stimulus (Trial 1 = 20.2 ± 4.8 mm), compared with DISPLACEMENT (Trial 1 = 6.8 ± 1.8 mm). Responses at the elbow varied considerably between the COMBINED and DISPLACEMENT stimuli. In particular, with the DISPLACEMENT stimuli an initial elbow flexion was replaced with a persistent elbow extension by the second trial. In contrast, the large initial elbow flexion with the COMBINED stimuli progressively decreased in amplitude with repeated stimuli, but an extension of the arm was not observed. The ANOVA indicated a significant Stimulus x Trial interaction ($F_{1,22} = 46.85$, $p < 0.0001$), as the amplitude of change in elbow angle converged with the later trials. The amount of elbow flexion with the first trial was significantly larger for the COMBINED stimulus (Trial 1 = 22.2 ± 2.4°), compared with DISPLACEMENT (Trial 1 = 2.6 ± 0.8°). The amplitude of the evoked burst of EMG activity in TA was generally larger with the COMBINED stimuli. The ANOVA did not identify a Stimulus x Trial interaction ($F_{130} = 3.92$, $p = 0.06$), whereas there were significant main effects of both Stimulus ($F_{130} = 13.26$, $p = 0.001$) and Trial ($F_{130} = 25.24$, $p < 0.0001$). The data for burst amplitude in AD demonstrate a clear Stimulus x Trial interaction ($F_{130} = 6.80$, $p = 0.01$) as the burst amplitudes with DISPLACEMENT are initially small (Trial 1 = 1.2 ± 0.3% MVC), but increase with later trials (Trials 8–10 = 3.9 ± 0.5% MVC). In contrast, the AD bursts evoked by the COMBINED stimuli show little adaptation with repeated trials, with an initial amplitude of 8.5 ± 2.9% MVC in Trial 1 then decreasing to 5.2 ± 1.2% MVC in Trials 8–10. Burst amplitude in SCM was larger with COMBINED stimuli than with DISPLACEMENT. A significant Stimulus x Trial interaction ($F_{126} = 12.26$, $p = 0.002$) was observed as the amplitude of the burst in SCM with the COMBINED stimuli decreased from an initial amplitude of 31.6 ± 6.7% MVC in Trial 1 to an amplitude of 10.5 ± 2.1% MVC in Trials 8–10, in contrast to what was observed with DISPLACEMENT, which did not change from near zero values for all trials. Bursts of activity in OO were seldom evoked with DISPLACEMENT, but were consistently evoked with the COMBINED stimulus. This yielded a significant main effect of Stimulus ($F_{130} = 25.59$, $p < 0.0001$), in the absence of a Stimulus x Trial interaction ($F_{130} = 1.46$, $p = 0.2$) or main effect of Trial ($F_{130} = 1.17$, $p = 0.3$).

### 3.4. Superimposition of Touch Displacement on Habituated Acoustic Startle Responses

Figure 7 shows data from one participant who was habituated to the STARTLE stimulus and then received DISPLACEMENT stimuli, before receiving the COMBINED stimuli. The first DISPLACEMENT trial (blue traces) resulted in a distinct forward sway observed in both the $COP_{AP}$ and Ankle traces, which was then not expressed with subsequent trials, consistent with the forward sway described in the preceding Results and previous studies [2,3]. However, with the reintroduction of the STARTLE the forward sway was again apparent with the first COMBINED trial (red traces) in the $COP_{AP}$ and Ankle, and with concomitant elbow flexion. Subsequent COMBINED trials reverted to the arm-tracking behaviour described earlier. This finding was not a unique occurrence and was consistently observed in all 8 participants who received this sequence of stimuli (Figure 8A). The EMG activity that accompanied these behaviours was qualitatively comparable to the pattern of adaptation observed when the STARTLE was superimposed on the habituated DISPLACEMENT stimuli (Figure 3A). That is, the first COMBINED stimulus resulted in a pronounced burst in TA (Trial 16 = 12.0 ± 3.0% MVC) and AD (Trial 16 = 18.5 ± 4.9% MVC), both of which decreased with repeated trials (Trial 20 = 4.6 ± 2.2% MVC and 7.9 ±% MVC, respectively). A complete report of the descriptive statistics and the result of the associated ANOVAs is provided in Table 3A.

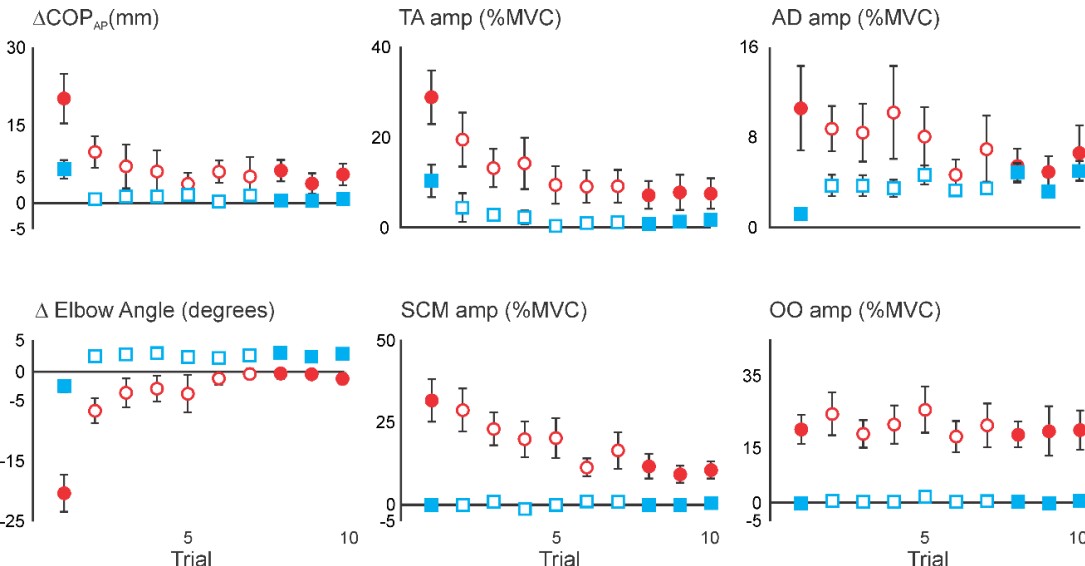

**Figure 6.** Trial-by-trial mean (SE) mechanical and muscle response amplitudes for all participants who received DISPLACEMENT (Blue squares) and COMBINED stimuli (Red circles) initially. Closed symbols indicate data points used in the statistical analysis. TA, tibialis anterior; AD, anterior deltoid; SCM, sternocleidomastoid; OO, orbicularis oculi; $COP_{AP}$, center of pressure anterior–posterior; MVC, maximum voluntary contraction.

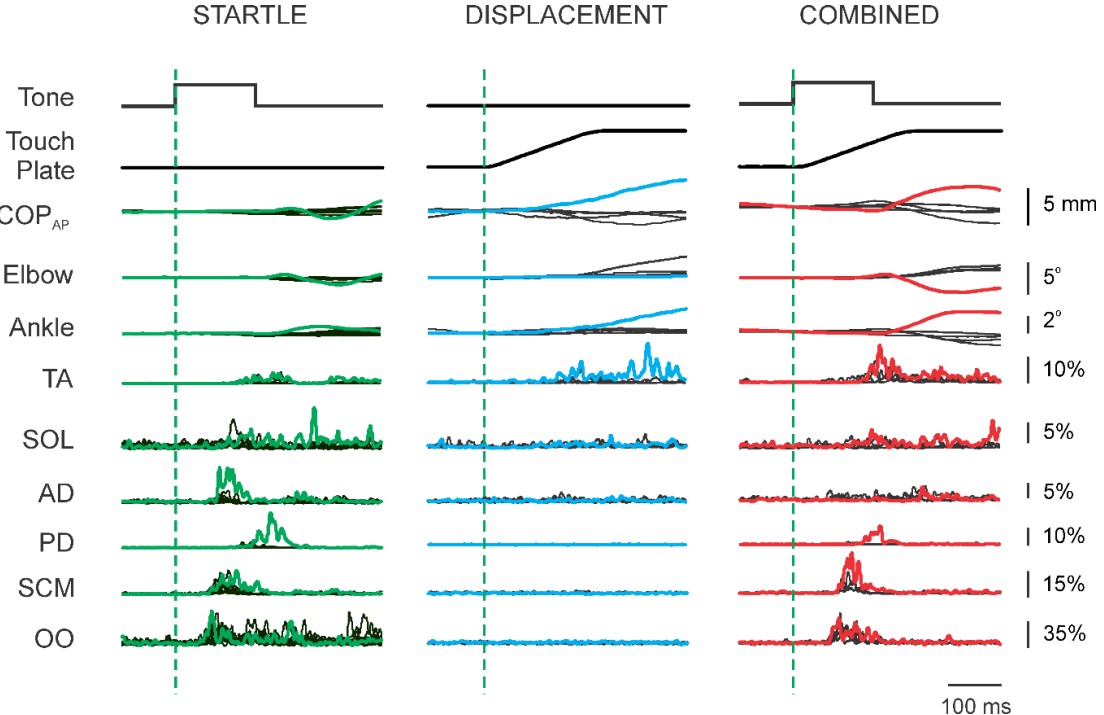

**Figure 7.** Sample data traces from a single participant who received the DISPLACEMENT stimuli after habituating to the STARTLE stimuli. Each cluster of traces represents the complete series of trials for that stimulus mode, with the coloured traces representing the first trial for that stimulus mode (Green, STARTLE; Blue, DISPLACEMENT; Red, COMBINED). The thin black lines of each cluster represent the subsequent trials. The vertical dashed line is aligned to the onset of the stimuli. Positive deflections in the $COP_{AP}$ traces represent forward, elbow traces represent extension, and ankle traces represent plantarflexion. $COP_{AP}$, center of pressure anterior–posterior; TA, tibialis anterior; SOL, soleus; AD, anterior deltoid; PD, posterior deltoid; SCM, sternocleidomastoid; OO, orbicularis oculi.

**Table 3.** Change in $COP_{AP}$ position and elbow angle, and normalized muscle response amplitudes.

| | Stimulus | | | | | | | Statistical Results | |
|---|---|---|---|---|---|---|---|---|---|
| A | $S_1$ | $S_{8-10}$ | $D_1$ | $D_5$ | $C_1$ | $C_5$ | $n$ | Main Effect | Tukey's HSD ($p < 0.05$) |
| **Mechanical** | | | | | | | | | |
| $\Delta COP_{AP}$ (mm) | 1.0 (2.7) | 0.8 (0.6) | 5.2 (1.1) | 1.0 (0.7) | 6.7 (2.4) | 3.4 (1.3) | 8 | 2.07 ($p = 0.093$) | |
| $\Delta$Elbow (deg) | 0.1 (1.2) | −0.2 (0.3) | −0.9 (0.3) | 2.0 (0.4) | −3.9 (1.4) | 3.3 (1.4) | 5 | 5.65 ($p = 0.002$) | $D_5 > C_1$, $C_1 < C_5$ |
| **EMG (%MVC)** | | | | | | | | | |
| TA | 2.1 (0.8) | 1.1 (0.4) | 4.2 (1.5) | 0.7 (0.5) | 12.0 (3.0) | 4.6 (2.2) | 8 | 8.03 ($p = 0.000$) | $D_5 < C_1$, $C_1 > C_5$ |
| AD | 5.1 (3.1) | 1.5 (0.6) | −0.2 (0.3) | 2.7 (0.7) | 18.5 (4.9) | 7.9 (2.1) | 7 | 8.78 ($p = 0.000$) | $D_5 < C_1$, $C_1 > C_5$ |
| SCM | 13.9 (4.4) | 7.5 (3.6) | −0.3 (0.5) | 0.5 (0.6) | 19.1 (6.8) | 11.1 (5.3) | 7 | 4.87 ($p = 0.002$) | $D_5 < C_1$ |
| OO | 21.1 (11.0) | 23.0 (9.8) | 2.1 (1.0) | 2.3 (1.1) | 27.6 (14.4) | 25.8 (10.6) | 8 | 3.29 ($p = 0.015$) | $D_5 < C_1$ |
| | Stimulus | | | | | | | Statistical Results | |
| B | $S_1$ | $S_{8-10}$ | $C_1$ | $C_5$ | $D_1$ | $D_5$ | $n$ | Main Effect | Tukey's HSD ($p < 0.05$) |
| **Mechanical** | | | | | | | | | |
| $\Delta COP_{AP}$ (mm) | 0.8 (0.6) | −0.1 (0.8) | 5.4 (0.7) | 0.1 (1.2) | 0.6 (1.0) | 0.5 (1.6) | 8 | 5.64 ($p = 0.001$) | $S_{8-10} < C_1$, $C_1 > C_5$ |
| $\Delta$Elbow (deg) | −0.1 (0.4) | −0.2 (0.2) | -2.0 (0.6) | 2.6 (1.0) | 1.2 (0.6) | 2.2 (1.0) | 5 | 6.57 ($p = 0.000$) | $C_1 < C_5$ |
| **EMG (%MVC)** | | | | | | | | | |
| TA | 2.8 (1.2) | 0.7 (0.4) | 6.9 (1.9) | 8.0 (2.6) | 0.9 (0.7) | 1.5 (1.3) | 7 | 5.27 ($p = 0.001$) | $S_{8-10} < C_1$, $C_5 > D_1$ |
| AD | 1.1 (0.4) | 0.9 (0.2) | 2.3 (1.6) | 3.3 (1.4) | 1.5 (0.5) | 1.3 (0.5) | 7 | 1.12 ($p = 0.373$) | |
| SCM | 8.4 (3.1) | 2.9 (1.5) | 5.0 (2.8) | 6.3 (3.0) | −0.1 (0.3) | −0.4 (0.3) | 7 | 2.74 ($p = 0.034$) | |
| OO | 11.3 (3.7) | 12.9 (5.0) | 18.7 (6.8) | 18.5 (5.7) | 0.5 (0.4) | 1.1 (0.8) | 7 | 5.61 ($p = 0.001$) | $C_5 > D_1$ |

Values are means (SE), $COP_{AP}$, center of pressure anterior–posterior; TA, tibialis anterior; AD, anterior deltoid; SCM, sternocleidomastoid; OO, orbicularis oculi; D, touch displacement; S, startle; C, combined; MVC, maximum voluntary contraction. Statistical results are $F$ statistics and $p$ values. Tukey's HSD applied to adjacent points in sequence only.

In Figure 8B, group averaged data are shown for participants who received the STARTLE initially, followed by the COMBINED stimuli. STARTLE did not result in a consistent impact on the $COP_{AP}$ and no demonstrable change in elbow angle, across participants. This was contrasted by bursts of EMG activity observed in each of the four muscles depicted. In TA, an initial burst (Trial 1 = 2.8 ± 1.2% MVC) decreased with repeated exposures (Trials 8–10 = 0.7 ± 0.4% MVC), whereas in AD a small initial burst (Trial 1 = 1.1 ± 0.4% MVC) showed little adaptation over repeated trials (Trials 8–10 = 0.9 ± 0.2 % MVC). A marked initial burst in SCM (Trial 1 = 8.4 ± 3.1% MVC) decreased with repeated trials (Trials 8–10 = 2.9 ± 1.5 % MVC). In contrast, pronounced initial bursts in OO (Trial 1 = 11.3 ± 3.7% MVC) were largely unchanged with repeated STARTLE exposure (Trials 8–10 = 12.9 ± 5.0% MVC). Subsequently, the DISPLACEMENT was superimposed with these habituated STARTLE responses. The initial COMBINED stimulus resulted in a forward sway (Trial 11 = 5.4 ± 0.7 mm), with an elbow flexion (Trial 11 = 2.0 ± 0.6°). With the subsequent COMBINED stimulus, the elbow flexion was replaced by an elbow extension, which remained largely unchanged (Trial 15 = 2.6 ± 1.0°). The forward sway in the $COP_{AP}$ progressively decreased with repeated COMBINED stimuli (Trial 15 = 0.1 ± 1.2 mm). COMBINED stimuli evoked increased burst amplitudes in all four muscles depicted, with no apparent adaptation in amplitude with repeated trials. Thereafter, DISPLACEMENT trials continued to induce elbow extension (Trial 16 = 1.2 ± 0.6°; Trial 20 = 2.2 ± 1.0°), with a concomitant small burst in AD (Trial 16 = 1.5 ± 0.5% MVC; Trial 20 = 1.3 ± 0.5% MVC). DISPLACEMENT did not consistently evoke activity in TA, SCM or OO. A complete report of the descriptive statistics and the result of the associated ANOVAs is provided in Table 3B.

## 3.5. Response Latencies

Response latencies could only be estimated for trials that elicited a clear response (see Methods). Thus, not all trials yielded a value and therefore the average value from a participant for each stimulus type was used in the analysis. The subsequent group averaged response onset latencies for each stimulus type are presented in Figure 9, for TA, AD, SCM and OO. Responses latencies were generally longest in TA and shortest in OO, reflecting the conduction distances from the stimulus source to the target muscle. The latencies in TA were consistent across all three stimulus conditions with times of 127.5 ± 14.6 ms, 125.4 ± 5.8 ms, and 126.3 ± 12.7 ms for DISPLACEMENT, STARTLE, and COMBINED, respectively ($F_{2,15} = 0.05$, $p = 0.96$). Responses in AD were also consistent across stimulus conditions with times of 104.9 ± 12.7 ms, 104.0 ± 5.3 ms, 99.1 ± 9.0 ms for DISPLACEMENT, STARTLE, and COMBINED, respectively ($F_{2,15} = 0.30$, $p = 0.74$). In contrast, response latencies in SCM and OO were significantly slower with DISPLACEMENT alone, than with STARTLE or COMBINED stimuli, whereas the response times in STARTLE and COMBINED were not different. For SCM, the response latencies were 107.6 ± 14.3 ms, 82.8 ± 4.9 ms, and 74.1 ± 8.5 ms for DISPLACEMENT, STARTLE, and COMBINED, respectively ($F_{2,15} = 6.15$, $p = 0.01$). For OO, the response latencies were 77.4 ± 17.1 ms, 47.5 ± 2.0 ms, and 48.4 ± 9.3 ms for DISPLACEMENT, STARTLE, and COMBINED, respectively ($F_{2,15} = 24.72$, $p < 0.001$).

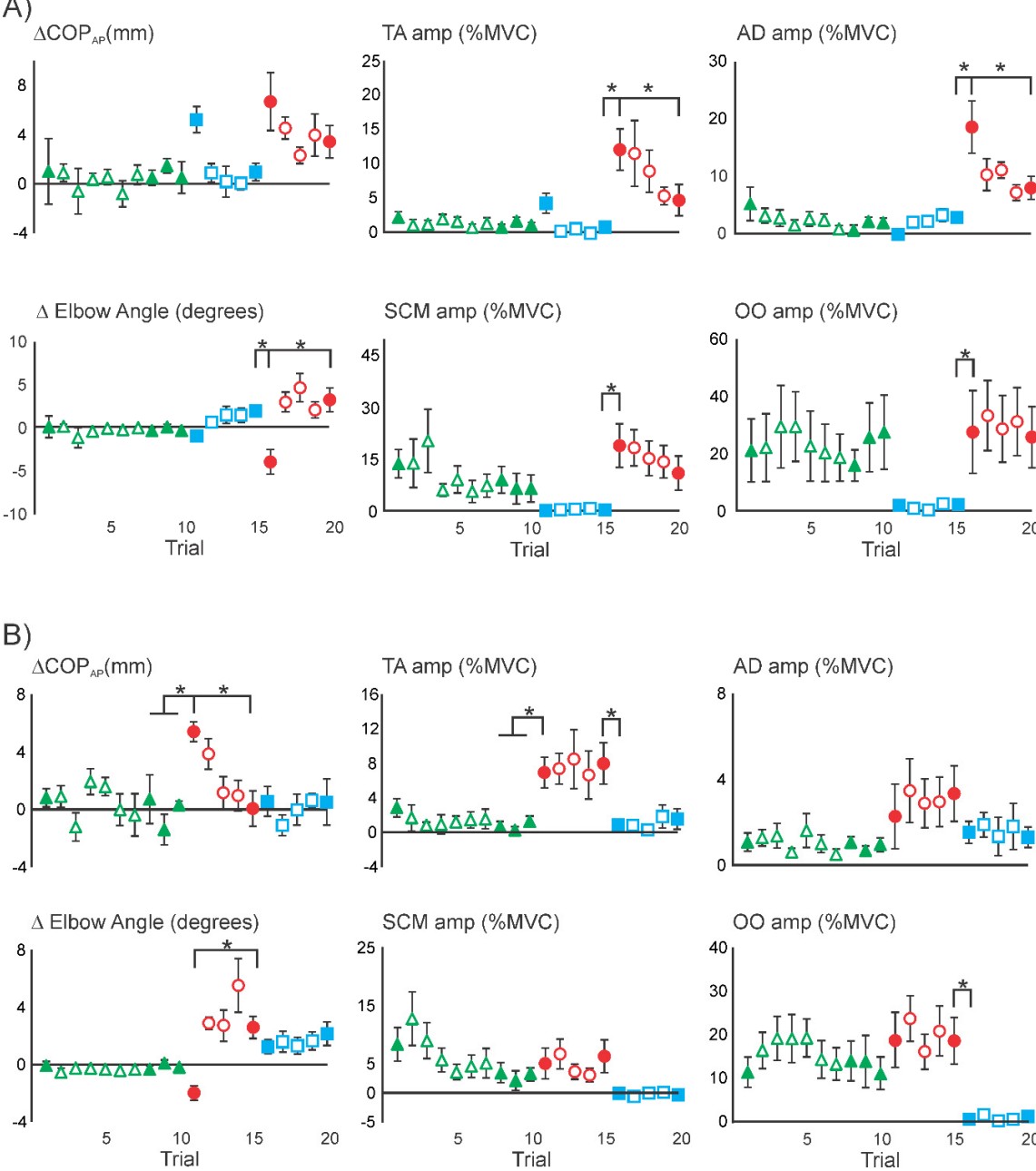

**Figure 8.** Trial-by-trial mean (SE) mechanical and muscle response amplitudes for participants who habituated to the STARTLE stimuli initially. (**A**) Participants subsequently received the DISPLACEMENT, followed by the COMBINED stimuli. (**B**) Participants subsequently received the COMBINED stimuli, followed by the DISPLACEMENT. Asterisks indicate differences identified by Tukey's HSD comparisons ($p < 0.05$). Blue squares, DISPLACEMENT; Red circles, COMBINED; Green triangles, STARTLE. Closed symbols indicate data points used in the statistical analysis. TA, tibialis anterior; AD, anterior deltoid; SCM, sternocleidomastoid; OO, orbicularis oculi; COP$_{AP}$, center of pressure anterior–posterior; MVC, maximum voluntary contraction.

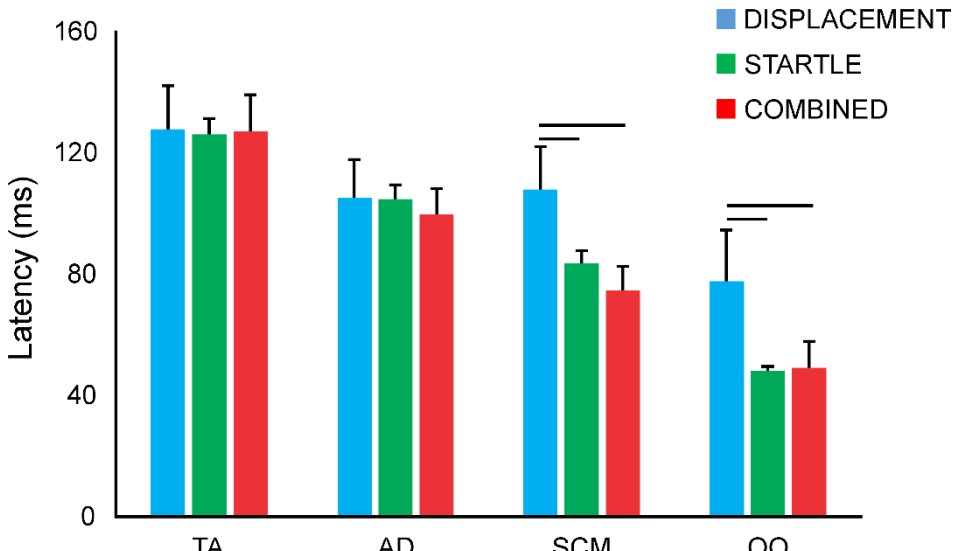

**Figure 9.** Mean (SE) onset latency for responses evoked in the respective muscle across all stimuli of the specific mode. Horizontal lines connect differences identified by Bonferroni adjusted, post hoc *t*-tests ($p < 0.05$). TA, tibialis anterior; AD, anterior deltoid; SCM, sternocleidomastoid; OO, orbicularis oculi.

## 4. Discussion

### 4.1. Interaction between Acoustic Startle and Habituated Touch Displacement Responses

Blouin et al. [12] demonstrated that introducing a STARTLE restored the amplitude of habituated neck postural responses in seated participants exposed to repeated accelerations of a sled. This finding strongly suggested that the initial response to the perturbation of the sled incorporated a postural response augmented by a superimposed startle response. Recently, we demonstrated that rapid displacement of a touch reference evoked a postural response on the first exposure, but an arm-tracking behaviour on subsequent exposures [2,3]. We speculated that the postural response evoked with the initial exposure might be related to the expression of a startle response and that the subsequent change in behaviour is related to the habituation of this startle. Contrary to our hypothesis, when we introduced a STARTLE after 10 trials of light touch displacement alone, the effect was to facilitate the arm-tracking behaviour that had been established, rather than restore the postural response (Figure 3A). However, when we introduced the STARTLE after only five trials of light touch displacement alone, the effect was to revert to the postural response initially (Figure 8A), supporting our hypothesis. These seemingly conflicting outcomes suggest that the interaction of the STARTLE with the reaction to the light touch displacement is dependent in part on the recent sensory history. In the first example, the facilitation of the arm-tracking behaviour suggests that the startle augmented a planned behavioural response, consistent with a StartReact-like effect. Whereas, in the second example, the restoration of the postural response suggests that the arm-tracking behaviour was not yet consolidated and that the startle biased the selection of the motor solution towards the balance corrective response or 'protective' option.

Startle is often observed to facilitate the execution of planned, voluntary movements [14]. The effect, referred to as the StartReact effect, has been observed with ballistic arm movements [15,16], ballistic ankle dorsiflexion [15], and stepping and obstacle avoidance [17], among other behaviours. Therefore, it was not entirely surprising that the arm-tracking behaviour in our study was facilitated when the STARTLE was co-presented with the touch plate displacement after 10 trials of touch plate displacement alone (Figure 3). The implication is that participants learned to preprogram the arm-tracking behaviour given the consistent direction and magnitude of the stimulus, which was then facilitated when the STARTLE was introduced. In stark contrast, when the STARTLE was co-presented with the touch plate displacement after only five trials, a postural response was evoked on the first COMBINED stimulus (Figure 8A, Trial 16). We note that the subsequent COMBINED stimuli (Trials 17–20) once

again, immediately evoked the arm-tracking behaviour, but were larger than the preceding touch plate DISPLACEMENT trials, suggestive of a StartReact-like effect. This suggests that there was still uncertainty as to the appropriate motor solution in response to the stimulus at the finger after only five exposures, and that STARTLE facilitated whichever solution was expressed. Furthermore, it is important to note that the switch in motor response, from the postural to the arm-tracking response, occurred in the presence of STARTLE, indicating that facilitation of the responses did not specifically interfere with the process of selecting a motor solution. The sudden switches, within a single trial, from a postural response to the arm-tracking behaviour observed here are consistent with the affordance competition hypothesis, wherein it is argued that multiple motor solutions are encoded in parallel before selecting the one that is implemented [18,19].

### 4.2. Interaction of Acoustic Startle and Novel Touch Displacement Responses

Combining a startle with the touch plate displacement on the first instance in naïve participants resulted in a substantially larger postural response than in participants who received the touch displacement alone initially (Figure 6). This finding is consistent with the StartReact-like effects described above and suggests the startle facilitated the postural response. It is important to note, however, that combining the stimuli in this manner evoked first trial postural responses in all 16 participants we tested in this way. This is very different from our previous findings where we have observed first trial postural responses in only about 60% of participants who receive a touch plate displacement alone [2,3]. In the present study, of the 16 participants who initially received touch displacements alone, only eight exhibited the postural reaction on the first trial. This suggests that the startle facilitated the release of the postural response in those participants who would not have otherwise reacted to the light touch displacement. The implication is that in the presence of a concomitant startling stimulus, ambiguous balance-related sensory cues are more likely to generate a balance response, even if the stimulus is misinterpreted as a threat and the resulting balance correction is unnecessary.

A second key finding from combining the startle with the touch displacement initially in naïve participants is that the arm-tracking behaviour did not emerge with repeated exposures to the COMBINED stimuli (Figures 4 and 5). One possible explanation might be that the effect of the startle on the facilitation of the postural response had not habituated sufficiently to allow for the expression of the alternate behaviour. Indeed, as shown in Figures 5 and 6, a prominent response in SCM, a common marker for the presence of a startle response [14,20,21], was still apparent after 10 trials. However, the presence of a startle response in itself is not enough to prevent the switching of behaviour from a postural response to the arm-tracking behaviour, as Figures 7 and 8 demonstrate. We propose that when the COMBINED stimuli are presented in the first instance (Figure 5, Trial 1), the startling nature of the acoustic stimulus biases the motor solution selection process towards a protective behaviour, that is, the postural response observed in this case. When combined with our findings that STARTLE facilitates established arm-tracking behaviour (COMBINED data in Figure 3) and facilitates both the postural response and arm-tracking when the motor solution is labile (COMBINED data in Figure 8), we suggest that STARTLE likely facilitates all alternative motor solutions, but with a bias towards protective solutions, such as the postural response, when the sensory information is ambiguous.

### 4.3. Technical Considerations

We interpret our findings of the first trial response to a touch displacement as including a startle component. In large part, this is based on the consistent evidence in the literature that unexpected stimuli, including balance disturbances, evoke startle responses [4,9]. Moreover, it has been demonstrated that combining startle-inducing stimuli leads to larger startle responses [4], consistent with our findings. However, our recordings of SCM did not consistently display a response to the touch displacement alone, even with the first exposure in naïve participants (Figure 3, Trial 1). Responses in SCM are commonly used as a marker of startle, and clear responses, that habituated as expected, were

apparent with the STARTLE in our study, which would support the use of SCM as a marker of startle. We suggest that in our study, the light touch displacement likely induced a weak startle, resulting in a small and inconsistent response in the SCM EMG recordings. This is corroborated in our study by the similarly inconsistent expression of the blink reflex, as indicated by the OO EMG recordings, in the DISPLACEMENT trials. The blink reflex is typically expressed as part of the startle response but is not generally considered a reliable marker of the startle response as it does not habituate [22]. Consistent with these previous findings, in our study, the STARTLE consistently evoked a blink reflex that did not habituate. The occasional, but inconsistent expression of the blink reflex and SCM response suggest that the light touch displacement was peri-threshold, or subthreshold, for the startle response. Previous work has also demonstrated that startle-like effects can be observed in the absence of SCM responses following a STARTLE [23–25]. Taken together, these findings suggest that startle responses need not be evident in SCM for unexpected stimuli to facilitate motor behaviours through a startle-like mechanism.

Whereas previous studies have shown interference between responses to stimuli delivered at different times (i.e., refractoriness, [26]), for our study, the purpose of the STARTLE was to provide a startling cue simultaneously with the onset of the touch displacement. We therefore used a 0 ms lag between the onset of the touch displacement and the STARTLE, which resulted in response latencies in TA and AD that did not differ across the three stimulus conditions (Figure 9). This was somewhat surprising as it has been demonstrated that the StartReact effect induces earlier onset reactions in voluntary reaction tasks (reviewed in [27]). The implication is that the postural responses evoked in TA and the arm-tracking responses evoked in AD might not be voluntary reactions per se, but perhaps are more automatic or reflexive, with less opportunity for reducing delays associated with the integrative processes within the neural chain.

This interpretation would be in conflict with the interpretation of Nonnekes et al. [23], who demonstrated that responses in TA to backward balance disturbances were quicker when accompanied by a STARTLE and suggested that the decrease in latency might be related to summative effects of medium latency postural responses and acoustic startle through a common relay in the reticular formation. Interestingly, Nonnekes et al. [23] also demonstrated that onset latencies of responses in gastrocnemius to forward balance perturbations were not influenced by the STARTLE, in contrast to the results in TA, which prompted the authors to suggest that the postural responses to forward and backward perturbations might be mediated via different neural circuits. In the present study, acoustic startle facilitated both the postural response (for example, Figure 5, Trial 1) and the arm-tracking response (for example, Figure 3 all COMBINED Trials) to light touch displacements. Moreover, the switching of responses, from postural to arm-tracking, was evident if the COMBINED stimulus occurred later in the protocol (for example, Figure 7), despite the apparent prevention of this behavioural switch if COMBINED was the initial stimulus (for example, Figure 6). Regardless, together these findings highlight the uncertainty surrounding the precise mechanism underlying the integration of startle with other sensorimotor circuitries and indicates that postural responses might arise, not from predetermined solutions to a sensory input, but from an ensemble of alternative motor solutions that are encoded simultaneously, and then expressed based upon the context and sensory state at the time of the disturbance [28,29].

*4.4. Functional Implications*

Unexpected sensory stimuli have been shown to generate stereotypical startle responses, regardless of the source or modality [4]. It has also been argued that the larger first trial responses to unexpected balance disturbances likely reflect the superimposition of a startle response on the underlying postural response [6,9,12]. Here, we demonstrate that overtly superimposing startle, by introducing an acoustic startle to the first occurrence of a light touch displacement, biases the evoked response towards a postural reaction, rather than the often observed arm-tracking response. This suggests that the functional outcome of the startling nature of unexpected stimuli might be to bias motor responses towards the most primitive, or protective interpretation of the stimulus. In this case, the slip detected

at the fingertip could indicate the touch reference moved relative to the finger or that the finger moved relative to the touch reference. However, in the presence of the acoustic startle, naïve participants uniformly and robustly reacted as though they had moved relative to the touch reference, inducing a forward corrective sway. Unexpected balance disturbances are often isolated, unique events with only a single opportunity to select the correct motor solution. Our results suggest that when multiple, unexpected stimuli are presented simultaneously, as might happen in authentic balance disturbances in natural contexts, the cumulative effect of the startle-like facilitation might serve to bias the selection of the motor response. That is, startle may not simply facilitate preprogrammed motor behaviours, as has been demonstrated previously, but may also constrain or limit the motor solution options in threatening situations.

One interpretation of these findings is that provision of a supplemental startling stimulus may serve to facilitate balance reactions. Indeed, the results of Blouin et al. [12], wherein habituated neck responses were restored to larger representations, would suggest that such a tactic might have functionally relevant benefits. However, caution should be applied to this interpretation as startle in the present study facilitated, and in some circumstances restored, an inappropriate postural response to the light touch displacement. These augmented "false-positive" postural reactions may themselves be destabilizing and fall-inducing. Nevertheless, it remains possible that combining appropriate augmented stimuli with a supplementary startle could induce functionally relevant postural reactions, which might be a useful approach to mitigate balance impairments in those with compromised sensory systems or sensorimotor integration. For example, embedding an auditory tone within a mobility device may serve to enhance reactions to unexpected instabilities.

**Supplementary Materials:** The following are available online at http://www.mdpi.com/2076-3417/10/1/382/s1. Spreadsheet S1: Misiaszek et al. 2019.xlsx, includes the complete dataset reported in this paper, including data that were not described in detail in the Results. Participant characteristics are also provided.

**Author Contributions:** J.E.M.: Conceptualization, funding acquisition, methodology, resources, software, validation, data curation, and writing—original draft. S.D.C.C.: Formal analysis, investigation, and writing—review and editing. A.J.M.: Formal analysis, investigation, and writing—review and editing. K.K.F.: Conceptualization, methodology, project administration, supervision, and writing—review and editing. All authors have read and agreed to the published version of the manuscript.

**Funding:** This work was supported by a grant from the Natural Sciences and Engineering Research Council (NSERC) Canada (RGPIN-2017-04175 to JEM) and an NSERC studentship to SDC.

**Acknowledgments:** The authors thank William Hodgetts (audiologist, Department of Communication Sciences and Disorders, University of Alberta) for assistance with developing and calibrating the acoustic stimulus.

**Conflicts of Interest:** The authors declare no conflict of interest.

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
