# Peer review of "Influence of Pairing Startling Acoustic Stimuli with Postural Responses Induced by Light Touch Displacement"

_applsci, doi:10.3390/app10010382_

Round 1

Reviewer 1 Report

The authors investigated how touch-induced balance reactions interact with startling acoustic stimuli.

The paper is well written and clear.

Only minor issues should be taken into account:

There are studies that underlying as motor control mechanisms (models) could take into account unexpected disturbances. As examples:

van de Kamp C, Gawthrop PJ, Gollee H, Loram ID (2013) Refractoriness in Sustained Visuo-Manual Control: Is the Refractory Duration Intrinsic or Does It Depend on External System Properties? PLoS Comput Biol 9(1): e1002843.    Quiet standing: The Single Inverted Pendulum model is not so bad after all P Morasso, A Cherif, J Zenzeri - PloS one, 2019   In the methods section a figure (or schema) of the setup could clarify the experiment   "COMBINED stimuli were presented with a 0 ms lag, given that the primary objective of this study was the interaction of STARTLE with DISPLACEMENT on responses related to activity in TA or AD"..How did you check the 0 ms lag?        

Could you please insert some speculation in the last section on how in your opinion technology should be modified (taking into account your results) to improve balancing?

some suggestion: Shima K, Shimatani K, Sato G, Sakata M, Giannoni P, Morasso P (2017) A Fundamental Study on How Holding a Helium‐filled Balloon Affects Stability in Human Standing. IEEE ICORR 2017, London UK, July 17-20, 1061-1066.

Author Response

We thank the reviewer for their feedback.

“There are studies that underlying as motor control mechanisms (models) could take into account unexpected disturbances. As examples: …”

We thank the reviewer for bringing to our attention these two papers.

The first, by van de Kamp et al., speaks to the concept of refractoriness in human motor control and specifically the potential for interference between two motor responses evoked by temporally displaced stimuli. Given that our study used a 0 ms lag between the onset of the two stimuli in the COMBINED condition, we are unable to provide insight that might differentiate between the serial ballistic hypothesis and the continuous optimal controller hypothesis. We have added content to the Technical Considerations section of our Discussion to touch on this.

The second paper, by Morasso et al., is a paper that will be of value to a current project ongoing in the laboratory. That a single inverted pendulum model is a functionally correct model to describe the motion of the body center of mass will be useful to assist in our interpretation of some interesting changes in sway patterns we are witnessing with sensory manipulations. However, it is unclear how this will contribute to the interpretation of the findings of the present study. In the present study, we evaluated brief episodic sway events, induced shortly after a specific stimulus. This outcome was the metric of interest, but we were not attempting to explain the mechanics for its appearance. We thank the reviewer for bringing this paper to our attention as it will inform our future work.
“… a figure (or schema) of the setup could clarify the experiment.”

A Figure has been added depicting the setup.
“…. How did you check the 0 ms lag?”

The stimuli were delivered by the computerized control program synchronously. This replicates the natural occurrence wherein an unexpected event would be expected to exhibit multiple sensory cues simultaneously. What we cannot know is if the stimuli were received by their sense organs, or transduced to their afferents, with 0 ms lag. We have clarified that the 0 ms lag refers to the stimulus delivery by the computerized program.
“Could you please insert some speculation in the last section on how in your opinion technology should be modified…”

We have added a comment suggesting how our results might be incorporated within mobility devices.
“Some suggestion:…”

Thank you for drawing our attention to this reference. It will be interesting to continue to follow this work as the balloon could be seen as providing a spatial reference, which then allows for the decreased sway observed. Alternatively, the floating balloon could be seen as an object that requires skilled control to stabilize and sway is then reduced to facilitate the skilled task (for example, Riley et al. 1999).

Reviewer 2 Report

In their earlier work the authors found that a person standing still on a foam surface holding a finger on piece of plastic will respond with a postural balance reaction when the piece is suddenly moved. This postural response, though, weakened on subsequent trials, so the assumption was made that the initial postural response is mostly due to a startling reflex. In a narrow sense, startling reflexes are triggered by loud and unexpected acoustic stimuli, but in a wider sense they can be triggered by a sudden burst of stimulation in any modality. Consequently, the authors now investigate the contribution of an auditory startle reflex to the described phenomenon of postural response. The manuscript is well written and organized, the hypotheses are motivated and adequately addressed by the experimental procedures. The experimental methods and the statistical evaluation are transparent, and the results are well presented in the text and in the figures. Although I consider the statistical evaluation adequate and carefully executed, I would suggest that the authors may look into using linear model estimation (LME) instead of a (repeated measures) ANOVA for their future investigations. The advantages of an LME are a better treatment of missing values, an integration of unbalanced designs, and a flexible mixing of within and between-subject variables. In the present study, for example, some participants had to be excluded from the statistical evaluation because their EMG dataset was not complete, or because their kinematic measurements were partially flawed. In an LME, in contrast, these subject would not have to be completely discarded and the clean part of their data could still be used, leading to an overall higher statistical power of the evaluation. In this respect, some just-not-significant results might have turned out significant, such as the interaction between Stimulus and Trial in section 3.3 on page 14 with a p-value of 0.06, which in turn would make the calculated main effects less reliable. But this is just a suggestion, and, as I said, the statistical treatment in the present manuscript is in my view perfectly fine and according to the standards. I have no suggestions for improvement, really, so I recommend the publication of the manuscript in its present form.

Author Response

We thank the reviewer for the kind words and for the suggestion of the linear model estimation. We will keep this in consideration in our future work.